# Bioanalytical Method Development, Validation and Stability Assessment of Xanthohumol in Rat Plasma

**DOI:** 10.3390/molecules27207117

**Published:** 2022-10-21

**Authors:** Vancha Harish, Waleed Hassan Almalki, Ahmed Alshehri, Abdulaziz Alzahrani, Sami I. Alzarea, Imran Kazmi, Monica Gulati, Devesh Tewari, Dinesh Kumar Chellappan, Gaurav Gupta, Kamal Dua, Sachin Kumar Singh

**Affiliations:** 1School of Pharmaceutical Sciences, Lovely Professional University, Jalandhar-Delhi G.T Road, Phagwara 144411, Punjab, India; 2Department of Pharmacology, Umm Al-Qura College of Pharmacy, Umm Al-Qura University, Makkah 21955, Saudi Arabia; 3Department of Pharmacology & Toxicology, Faculty of Pharmacy, Northern Border University, Rafha 91911, Saudi Arabia; 4Pharmaceuticals Chemistry Department, Faculty of Clinical Pharmacy, Al Baha University, Al Baha 65779, Saudi Arabia; 5Department of Pharmacology, College of Pharmacy, Jouf University, Sakaka 72341, Saudi Arabia; 6Department of Biochemistry, Faculty of Science, King Abdulaziz University, Jeddah 21589, Saudi Arabia; 7Faculty of Health, Australian Research Centre in Complementary & Integrative Medicine, University of Technology Sydney, Ultimo, NSW 2007, Australia; 8Department of Pharmacognosy and Phytochemistry, School of Pharmaceutical Sciences, Delhi Pharmaceutical Sciences and Research University, New Delhi 110017, Delhi, India; 9School of Pharmacy, International Medical University, Bukit Jalil, Kuala Lumpur 57000, Malaysia; 10School of Pharmacy, Suresh Gyan Vihar University, Mahal Road, Jagatpura, Jaipur 302017, Rajasthan, India; 11Department of Pharmacology, Saveetha Dental College and Hospitals, Saveetha Institute of Medical and Technical Sciences, Saveetha University, Chennai 602105, Tamil Nadu, India; 12Uttaranchal Institute of Pharmaceutical Sciences, Uttaranchal University, Dehradun 248007, Uttarakhand, India; 13Discipline of Pharmacy, Graduate School of Health, University of Technology Sydney, Ultimo, NSW 2007, Australia

**Keywords:** xanthohumol, rat-plasma, bioanalytical method, RP-HPLC, stability

## Abstract

Xanthohumol (XH) a prenylated chalcone has diverse therapeutic effects against various diseases. In the present study, a bioanalytical method was developed for XH in rat plasma using reverse phase high performance liquid chromatography. The validation of the method was performed as per ICH M10 guidelines using curcumin as an internal standard. The Isocratic elution method was used with a run time of 10 min, wherein the mobile phase ratio 0.1% *v*/*v* OPA (A): Methanol (B) was 15:85 *v*/*v* at flow rate 0.8 mL/min and injection volume of 20 µL. The chromatograms of XH and curcumin was recorded at a wavelength of 370 nm. The retention time for XH and curcumin was 7.4 and 5.8 min, respectively. The spiked XH from plasma was extracted by the protein precipitation method. The developed method was linear with R^2^ value of 0.9996 over a concentration range of 50–250 ng/mL along with LLOQ. The results of all the validation parameters are found to be within the accepted limits with %RSD value less than 2 and the percentage recovery was found to be greater than 95%. Based on the %RSD and percentage recovery results it was confirmed that the method was precise and accurate among the study replicates. LOD and LOQ values in plasma samples were found to be 8.49 ng/mL and 25.73 ng/mL, respectively. The stability studies like freeze thaw, short term and long-term stability studies were also performed, %RSD and percentage recovery of the XH from plasma samples were within the acceptable limits. Therefore, the developed bioanalytical method can be used effectively for estimation of XH in plasma samples.

## 1. Introduction

The bioactive chemicals found in beer are thought to be responsible for the tangible impact of the beverage. Xanthohumol (XH) notably, has grabbed scientists’ interest due to its variety of biological properties as it is used as a bittering agent in manufacturing of beer [1,2]. XH is a prenylated chalcone obtained from the female blossoming of Hops plant belongs to cannabaceae family. Currently it is a molecule of great interest because of its wide variety of biological potentials such as hot flushes, toothache, cancer, glioblastoma, excitability, osteoporosis, diabetes, food supplement, restlessness, anti-ageing agent, inflammation, digestion enhancement, neuralgia, earache, antiatherogenic and it is also used in traditional medicine (insomnia and mental stress) [2,3]. The chemical structure of XH is shown in Figure 1a. Due to its high therapeutic potential towards various deadly diseases, XH is evolved as a novel molecule of interest for current researchers [4]. XH is considered as a broad spectrum anticarcinogenic agent due its excellent capacity to inhibit all the stages of cancer by acting on procarcinogen activation enzymes and detoxifying enzymes of cancer [5,6]. Because of this good pharmacological properties XH could promote to clinical translation. During developmental stages of XH as clinically approved drug for various diseases, analytical method will play a crucial role for the estimation of drug concentration in both in vitro and in vivo environments. They are also important during the characterization of various biopharmaceutical parameters (dissolution, permeation and pharmacokinetics). Therefore, the present research aimed to develop economical robust bioanalytical method for estimation of XH in biological environment (rat plasma).

Nowadays, researchers are focusing on the evaluation of pharmacokinetic parameters of the XH in humans and animals. The quantification of XH in biological samples require simple, efficient, analytical methods. So, there is immediate need to develop the bioanalytical method to evaluate the biological samples of humans or animals. However, the analytical methods available are costlier (i.e., LC/MS-MS) and it is not feasible for the low funding laboratories to establish them. There are very few methods reported, out of which most of them are developed for the estimation of mixture of prenylated chalcones, but not single XH and are not in biological samples. For best of our knowledge, research group of Avula et al., 2004 has developed and validated HPLC method to determine the XH in rat plasma, fecal and urine samples after giving XH through intravenous route (20 mg/kg) and oral route (50, 100, 200, 400 and 500 mg/kg) body weight [7]. Avula’s group used gradient elution method for separation using mobile phase (A: 0.025% trifluoroacetic acid in water, B: 0.025% trifluoroacetic acid in acetonitrile) at a flow rate of 1 mL/min. The conditions for gradient elution were 65% A and 35% B for 25 min and later 25% A and 75% B followed by washing for 10 min after every run with 5% A and 95% B [7]. However, retention time (Rt) of XH by Avula’s method was above 15 min and XH was identified at different Rt in different samples. In another study, Sus et al., 2018 developed and validated a reverse phase HPLC method for the quantification of prenylated chalcones (XH) and prenylated flavones (6-prenylnaringenin and 8-prenyl naringenin) in rat plasma and urine [8]. However, they used extract of hops and used binary gradient method for elution and separation containing mobile phase (A: 5% formic acid in water; B: acetonitrile containing 5% formic acid and 10% water) at a flow rate of 1 mL/min. The method was operated as follows, for 1 min 100% A to 70% A, hold at 75% A for 3 min, decreasing A to 25% within 12 min and to 0% in 14 min followed by equilibrium for 20 min at 70:30. The Rt of XH identification was after 15 min [8]. In another study Nowak et al., 2020 investigated the pharmacokinetic profile of XH in pure form, prenylflavonoid extract and spent hops in rat plasma. For the estimation of XH they used HPLC method, gradient elution was used and is operated as follows solvent B (1% formic acid in methyl cyanide) 40–100% in solvent A (1% formic acid in water) for 15 min at 0.8 mL/min flow rate, after first 2 min at 40% B and then from 100–40% B for 7 min and held for 2 min in 40% B. XH was identified at 14.598 min [9]. Using the above two methods the quantification of XH is time consuming (Rt is near to 15 min or above 15 min) and gradient elution method is somewhat complicated to run. Stevens et al., 1999 has developed HPLC-tandem mass spectrometry method to quantify XH, isoxanthohumol. Desmethylxanthohumol, 6- and 8-prenylnaringenins and 6-geranylnaringenin from beer and crude methanolic extracts, this method mainly offers direct analysis of crude extracts and beer [10]. Vazquez et al., 2019 developed HPLC-DAD-MS/MS method to determine XH in hops, beer and food supplements [11]. All the above discussed methods are expensive and require more time for analyzing the samples. Therefore, our current research focused on the development and validation of the simple, robust, cost effective, and less time-consuming laboratory scale RP-HPLC bioanalytical method by using UFLC according to ICH M10 guidelines. The method was developed based on isocratic elution, which is easier to operate, and the pressure of the system will be constant when compared to the gradient elution. Curcumin (Figure 1b) was used as internal standard in the present study. The validation parameters like specificity, accuracy, limit of detection (LOD) and lower limit of quantification (LLOQ), linearity and range, precision, stability and system suitability were evaluated for confirming the successful validation. The applicability of the developed and validated method was in study of pharmacokinetic profile, bioavailability of XH in biological samples.

## 2. Results and Discussion

The bioanalytical method for the estimation of XH in rat plasma was developed as per ICH M10 guidelines. To develop the method, preliminary investigations were carried out in compliance with protocols published in the literature and pharmacopoeias. Many attempts were made with varying composition of the mobile phase containing solvents (acetonitrile, methanol), and water containing OPA (0.1%, 0.5%, 1% and 2% *v*/*v*), flow rate (0.8, 1.0, and 1.2 mL/min), to improve the efficiency of the method and for the least peak tailing and better peak resolution. Several attempts have failed to give better peak resolution, tailing factor, theoretical plates and less retention time. The composition of OPA and acetonitrile has not given any peak. Lastly, methanol and water containing 0.1% *v*/*v* OPA has given peak. Later, optimization was performed with varied composition of methanol, 0.1% *v*/*v* OPA and flow rate to identify the peak with better resolution, less retention time, theoretical plates and least peak tailing. Therefore, finally, with the mobile phase ratio 0.1% *v*/*v* OPA (A): Methanol (B) was 15:85 *v*/*v* at flow rate 0.8 mL/min has given peak with better resolution, less retention time, theoretical plates and least peak tailing.

### 2.1. Chromatograms of Mixture Containing Curcumin and XH

The chromatogram of curcumin and XH mixture in plasma is shown in Figure 2a. The retention time of curcumin and XH were found to be 5.819 and 7.420 min, respectively.

### 2.2. Specificity Studies

The absence of peak at the retention time of curcumin and XH revealed that the plasma matrix had no effect on the quantitative estimation of XH and curcumin extracted from the blood. The chromatogram of blank plasma was shown in Figure 2a. Similarly, when XH at a concentration of 200 ng/mL and curcumin at 1 µg/mL were spiked with plasma, showed absence of any peak of plasma on the retention times of both drugs curcumin and XH (Figure 2b).

### 2.3. Development of Calibration Curve (Linearity and Range)

The calibration curve for XH prepared in plasma indicated excellent linearity with R2 value of 0.9995 (Figure 3).

### 2.4. Matrix Effect

In order evaluate the matrix effect, the LLOQ and HQC samples were injected six times each with HPLC and the percentage recovery and %RSD was calculated. The percentage recovery was found to be above 95% in all cases indicating their no interference of any other unidentified compound in the sample. It also confined by the %RSD less than 2 (Table 1). Therefore, it was concluded that there was absence of a matrix effect of analyte with the plasma matrix and can be used for bioanalytical method development and validation as the results lies in acceptable limits, i.e., ±15%.

### 2.5. Accuracy Studies

The percentage absolute recovery of XH in the plasma is used to calculate the accuracy. The percentage recovery of XH in plasma was found to be greater than 95% implying that the developed method is highly accurate (Table 2). It is important to note that in each instance, the %RSD was less than 2, showing that the results were reproducible.

### 2.6. Precision Studies

The intraday, interday and inter-analyst precision (Table 3) studies were carried out for XH in plasma. The %RSD of the areas recorded for LLOQ, LQC, MQC and HQC samples was found to be less than 2 indicating precision of the method.

### 2.7. LOD and LLOQ

The LOD and LLOQ for XH in plasma samples were found to be 8.49 ng/mL and 25.73 ng/mL, respectively. These indicated that the method was sensitive for detection of both the drugs at lower concentrations.

### 2.8. Carry-Over and Dilution Integrity

There was no identification of carryover (analyte peak) after injecting the blank sample spiked with internal standard. The dilution integrity was also present in the acceptable limits by giving percentage drug recovery greater than 95% and %RSD less than 2. Therefore, the samples will not affect the accuracy and precision.

### 2.9. System Suitability

The number of theoretical plates were always >2000 in all chromatographic runs to ensure good column efficiency throughout the developed separation process. The tailing factor for XH peak never exceeded 2 in all peaks demonstrating good peak regularity (acceptance limits < 2). The results were shown in Table 4.

### 2.10. Stability Study of Plasma Samples

Freeze thaw (Table 5), short term (Table 6) and long-term stability (Table 7) studies were conducted for LQC, MQC and HQC plasma samples. The results indicated more than 95% recovery of XH in all cases as well as %RSD less than 2. The outcome of these studies indicated the stress as well as long term storage stability of drugs in plasma samples.

## 3. Materials and Methods

### 3.1. Materials

#### 3.1.1. Chemicals and Equipment

XH was obtained as a gift sample from Siemen. Hop Steiner, Germany. Curcumin was purchased from HiMedia (Mumbai, India). HPLC grade Methanol, Ortho-Phosphoric acid were purchased from Merck (Mumbai, India). Triple distilled water was used throughout the study. UFLC system (Schimadzu LC-20AD, Tokyo, Japan) with a photodiode array detector (SPD-M20A, Tokyo, Japan) and a Rheodyne sample injector loop (20 µL) was used for quantitative analysis. Vortex mixer and cooling centrifuge were from REMI, India.

#### 3.1.2. Animals

Six male Sprauge Dawley rats were purchased from National Institute of Pharmaceutical Education and Research (NIPER), Mohali, India for the present study. The age of all rats was between 10 and 11 weeks and weight in the range of 250–350 g. The rats were kept in polypropylene cages lined with husk under the temperature of 25 ± 2 °C; relative humidity of 55 ± 10% and 12:12 light: dark cycle. The animals were fed with standard pellet diet and water and libitum. The study protocol was approved by the Institutional Animal Ethics Committee of School of Pharmaceutical Sciences, Lovely professional University (Protocol no: LPU/IAEC/2021/88).

### 3.2. Methods

#### 3.2.1. Chromatographic Conditions

Bioanalytical method was developed as per ICH M10 guidelines for the quantification The Isocratic elution method was used with run time of 10 min, wherein, the mobile phase ratio 0.1% *v*/*v* OPA (A): Methanol (B) was 15:85 *v*/*v* at flow rate 0.8 mL/min and injection volume of 20 µL of XH in rat plasma. Nucleodur reverse phase C18 column with dimensions 250 mm × 4.6 mm i.d., 5 µ, macherey nagel was used for plasma component separation. The XH chromatogram was detected at 370 nm. LC solution software, version 12.1 is used to operate the whole system.

#### 3.2.2. Collection of Blood and Extraction of Plasma

The blood was collected in radioimmunoassay (RIA) vials containing ethylene diamine tetra acetic acid (EDTA) crystals from rats through retro orbital puncture using capillary tubes. The rat was immobilized, the neck was gently scruffed, and one of the eyes was forced to protrude. A capillary tube with smooth radial surface was introduced anteriorly into the retro-orbital plexus of rat’s eye. The blood was allowed to drain into the RIA vial containing EDTA by capillary action through capillary tube. After gentle mixing, the blood-filled RIA vials were centrifuged for 15 min at 10,000 rpm. A micropipette was used to collect the clear supernatant which was then stored in deep freezer at −20 °C for subsequent processing.

#### 3.2.3. Preparation of Blank Plasma

Blank plasma was prepared by adding 2 mL of methanol: ACN mixture (1:1 *v*/*v*) to 1 mL plasma. The resultant mixture was vortexed (5 min) followed by centrifugation (15 min, 10,000 rpm) in order to precipitate and separate the plasma proteins. The clear supernatant obtained after centrifugation is collected in a clean 100 mL volumetric flask and the volume was made up to 100 mL by methanol.

#### 3.2.4. Preparation of Standard Stock Solution

The standard stock solution containing 100 µg/mL concentration of XH was prepared with plasma in methanol. Standard stock solution of XH was prepared by adding 10 mg of XH to 2 mL of plasma and vortexed for 10 min to obtain spiked mixture of plasma and XH. To this spiked mixture, 2 mL of methanol and ACN (1:1 *v*/*v*) was added and vortexed again for 5 min in order to precipitate the plasma proteins. Then, the resultant mixture was centrifuged for 15 min at 10,000 rpm and 4 °C, supernatant was collected with micropipette in to a 100 mL volumetric flask. The volume was made up to 100 mL with methanol to obtain stock solution with 100 µg/mL concentration (stock solution A). Furtherly, serial dilutions were prepared in order to get the solutions of concentration 10 µg/mL (stock solution B) and 1 µg/mL (stock solution C) solutions.

#### 3.2.5. Preparation of Internal Standard (IS)

Curcumin (10 µg/mL) is used as the IS in the present study. In order to obtain the required concentration of curcumin solution, 10 mg of curcumin was taken in to 100 mL volumetric flask containing 20 mL of methanol and sonicated for 5 min for complete dissolution of curcumin. The volume was made up to 100 mL with methanol to obtain 100 µg/mL stock solution. Furtherly, 1 mL of 100 µg/mL solution was taken and diluted to 10 mL to get 10 µg/mL solution which is used for preparation of dilutions.

#### 3.2.6. Method Specificity

Specificity of the method was investigated to observe the possible interactions with the peaks of the blank plasma and drug spiked plasma. For this blank plasma dilution (2 µg/mL) was prepared by diluting 0.2 mL of 100 µg/mL blank plasma to 10 mL. From 2 µg/mL solution 200 ng/mL solution was prepared. Then, the solution containing 200 ng/mL of XH and 1 µg/mL curcumin concentration was prepared by taking 2 mL of 10 µg/mL XH solution (stock solution B) and 1 mL of 10 µg/mL concentration curcumin solution, the final volume was made up to 10 mL with methanol. The resultant solution contains 2 µg/mL concentration of XH and 1 µg/mL concentration of curcumin. 20 µL of blank plasma solution and the solution containing XH and curcumin (IS) were injected to the HPLC system and analysed for peak interactions at 370 nm [12,13].

#### 3.2.7. Development of Calibration Curve

Aliquots in the range 50–250 ng/mL were prepared by taking 0.5 mL, 1.0 mL, 1.5 mL, 2.0 mL and 2.5 mL of stock solution C (10 µg/mL) in separate volumetric flasks of volume 10 mL. During the development of calibration curve 25.73 ng/mL is taken as LLOQ and 250 ng/mL is taken as upper limit of quantification. To all these volumetric flasks 1 mL of 10 µg/mL of curcumin solution was spiked as IS and sonicated for 5 min to ensure the homogenous mixing, volume was made up to 10 mL with methanol. The resultant mixtures contain 1 µg/mL concentration of IS in all the aliquots. Initially IS was also spiked to blank and injected to HPLC before injecting the aliquots containing drug. All the aliquots were injected to HPLC (n = 5) and mean area, retention time of the chromatograms (IS and XH in plasma) were recorded at 370 nm, respectively. Calibration curve was plotted by taking concentration (ng/mL) on *X*-axis and area(mAu) on *Y*-axis. Linearity of the analytical method was calculated by determining the correlation coefficient, slope and intercept of the curve [12,13].

#### 3.2.8. Method Validation

The validation of the developed method was performed as per ICH M10 guidelines. The validation parameters like linearity and range, accuracy, precision, LOD and LOQ were evaluated. The performance of the system was studied by evaluating system suitability parameters such as height equivalent to theoretical plate (HETP), tailing factor, theoretical plates, LOD and LOQ [ICH, M10, 2019].

#### 3.2.9. Linearity and Range

The calibration curve was plotted against the ratio of analyte concentration and IS concentration versus the ratio of peak area of analyte and peak area of internal standard. Using MS-Excel software, the slope, intercept and regression coefficient were calculated and reported.

#### 3.2.10. Matrix Effect

Matrix effect is defined as the change in the analyte response due to interfering and undetectable components in the sample matrix. Matrix effect is mainly due to alteration in the ionization efficiency. Matrix effect is basically evaluated by analyzing LLOQ and HQC samples (n = 6) prepared using plasma matrix. The accuracy and precision should be in between ±15%.

#### 3.2.11. Accuracy

The accuracy study was performed based on the percentage absolute recovery of XH from four quality control samples viz lower limit of quantification (25.73 ng/mL, LLOQ), lower quantifiable concentration (80%, LQC), middle quantifiable concentration (100%, MQC) and high quantifiable concentration (120%, LQC) of the middle concentration of the method, i.e., 150 ng/mL. LQC, MQC and HQC solutions were prepared by taking 1.2 mL, 1.5 mL and 1.8 mL, respectively from stock solution C in to separate 10 mL volumetric flasks. To all the quality control samples, 1 mL of 10 µg/mL concentration curcumin solution is added as IS, so as to obtain the final concentration 1 µg/mL. Percentage absolute recovery of XH from LLOQ, LQC, MQC and HQC was calculated by the formula mentioned in Equation (1) and is used to calculate the accuracy of the method. Each concentration was analysed six times (n = 6) and mean, standard deviation (SD), percentage relative standard deviation (%RSD) were calculated to confirm the accuracy.
(1)$$Percentage\;absolute\;recovery=\frac{Actual\;concentration\;recovered}{Theoritical\;concentration}\times 100$$

#### 3.2.12. Precision Studies

Precision studies were performed with LLOQ, LQC, MQC and HQC samples to evaluate the consistency of results between many samples of same concentration on the same day and on other days of analysis. Three types of precision studies can be investigated namely, intraday (repeatability), interday (reproducibility and inter-analyst precision. These studies can be performed by injecting (n = 6) quality control samples on same day and three consecutive days under same experimental conditions. Inter analyst precision was performed by injecting the samples on the same day by three different analysts. Their mean, SD, %RSD were calculated [12].

#### 3.2.13. Determination of LOD and LLOQ

LOD and LLOQ for XH spiked in plasma were calculated based on SD of intercept and slope of linear regression equation using Equations (2a) and (2b) [12, ICH M10, 2019].
(2a)$$\mathrm{LOD}=\frac{\left(3.3\right)\mathrm{SD}}{S}$$
(2b)$$\mathrm{LLOQ}=\frac{\left(10\right)\mathrm{SD}}{S}$$
where SD is the standard deviation of the intercept, S is slope of the linearity curve.

#### 3.2.14. Carry-Over

Carry-over study is performed in order to identify the presence of small sample peak after injecting a blank following the analyte injection that gives big peak. The presence of a small, unexpected peak of analyte with blank is the indication of carryover. In the present study, carry-over is analyzed by injecting the blank after every injection of HQC. Carry-over should not be greater than 20% of LLOQ analyte peak and 5% than IS.

#### 3.2.15. Dilution Integrity

Dilution integrity was analyzed by combining the matrix and the analyte at a concentration over HQC and diluting the sample with blank matrix. The accuracy and precision of the dilution should not exceed ±15% in order to meet the requirements.

#### 3.2.16. System Suitability

The system suitability study was performed by injecting six times the lowest concentration among the concentrations of the calibration curve (i.e., 50 ng/mL). The parameters such as tailing factor, theoretical plates, HETP, LOD and LOQ were calculated for the obtained chromatogram.

#### 3.2.17. Stability Study of XH Spiked Plasma Samples

The XH spiked plasma samples were exposed to three freeze–thaw cycles, short term stability at ambient temperature for 3 h and long-term stability at −20 °C for 3 weeks. To perform freeze thaw stability study of XH in plasma, plasma (3 mL) was taken in a RIA vial and 3 mg of XH was added to get the concentration of 1000 µg/mL. The resultant mixture was vortexed for 5 min and kept in a deep freezer at −20 °C for 12 h. After the sample had frozen the vial was removed and allowed to thaw at normal temperature. 1 mL plasma was withdrawn from the thawed samples (cycle 1) using micropipette. After 12 h the remaining 2 mL sample was again kept in the deep freezer for second cycle. The collected 1 mL plasma containing XH was precipitated and centrifuged. The clear supernatant collected was diluted with 100 mL of methanol in order to get the concentration of 100 µg/mL. From this LLOQ (25.73 ng/mL), LQC (120 ng/mL), MQC (150 ng/mL) and HQC (180 ng/mL) samples were prepared. In the similar manner as cycle 1, with the remaining 2 mL sample the freeze–thaw cycle 2 and cycle 3 were carried out and LLOQ, LQC, MQC and HQC samples were prepared. IS (curcumin) was added to all the samples in order to obtain the concentration of 1 µg/mL. All the dilutions from cycle 1, cycle 2 and cycle 3 were prepared in triplicate and injected to HPLC and analyzed at 370 nm. The mean. SD, %RSD were calculated for each concentration [12,14,15].

The short-term stability studies of plasma spiked with XH were determined at room temperature after 1 h, 2 h and 3 h before extraction. For short term stability, 1000 µg/mL solution was prepared by adding 3 mg of XH to RIA vial containing 3 mL of plasma and vortexed for 5 min. The RIA vial containing XH spiked plasma was kept at room temperature. After 1 h, 2 h and 3 h sample (1 h) was withdrawn and extracted the drugs from the plasma. After extraction of drugs, the sample is processed for the preparation of LQC, MQC and HQC samples after each interval. For all these quality control samples IS was added so as to get the concentration of 1 µg/mL. Samples were prepared in triplicate and injected to HPLC system and analyzed at their respective retention times at 370 nm. The mean. SD and %RSD was calculated for each concentration [16]. For long term stability studies, three RIA vials were taken and 1 mL of plasma and 1 mg of XH was added to each vial. The concentration of drug in each vial is 1000 µg/mL concentration. All the three vials were vortexed for 5 min and kept in deep freezer at −20 °C. The vials were taken out from freezer after 1, 2 and 3 weeks, respectively. After each interval, the drugs were extracted from plasma and processed for preparation of LQC, MQC and HQC samples and IS (1 µg/mL) was added. All dilutions were produced in triples and analyzed their retention times at 370 nm by injecting in to HPLC system. For each concentration, the mean, SD and %RSD were computed [9,12].

#### 3.2.18. Statistical Analysis

Analysis of variance (ANOVA) was assessed for the results presented in the study using Graph Pad Prism version 7.0 (GraphPad software Inc., CA, USA). After the ANOVA assessment, Turkey’s multiple comparisons test was applied and mean ± SD were recorded.

## 4. Conclusions

The present demonstrated the effective development of RP-HPLC bioanalytical method for quantification of XH in rat plasma. The developed method has demonstrated great sensitivity, accuracy and precision indicating that it is repeatable. This method can be used in laboratory level to estimate the plasma concentration of XH. Over comparing to the other methods developed by various researchers, current method is less time consuming and not as complicated. The major advantage of this method was cost effective. In addition, this method has set a precedent for analysts to use it in variety of in vivo investigations, including pharmacokinetic, drug distribution, plasma protein binding and drug metabolism studies.

## Figures and Tables

**Figure 1 molecules-27-07117-f001:**
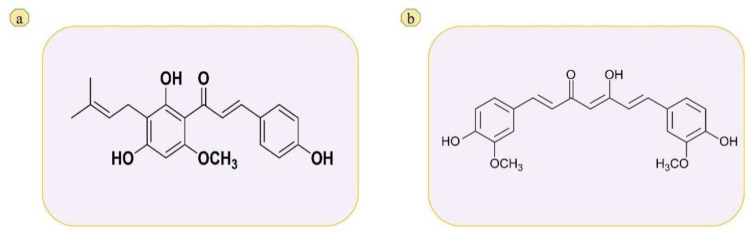
Chemical structure of (**a**) XH and (**b**) curcumin.

**Figure 2 molecules-27-07117-f002:**
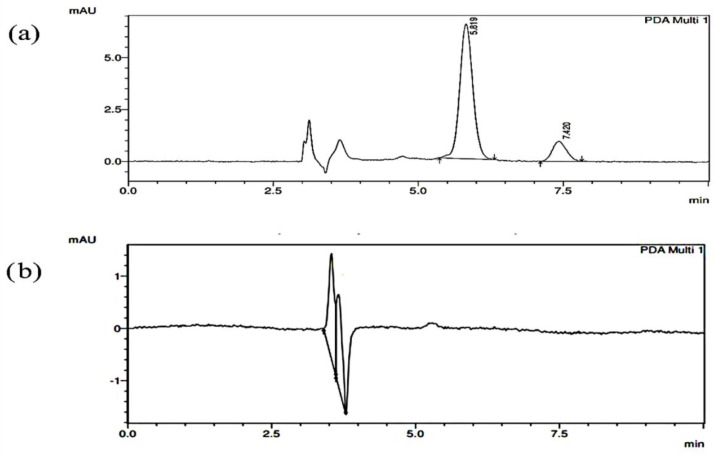
Representing chromatogram of (**a**) plasma spiked with IS and XH. (**b**) blank plasma.

**Figure 3 molecules-27-07117-f003:**
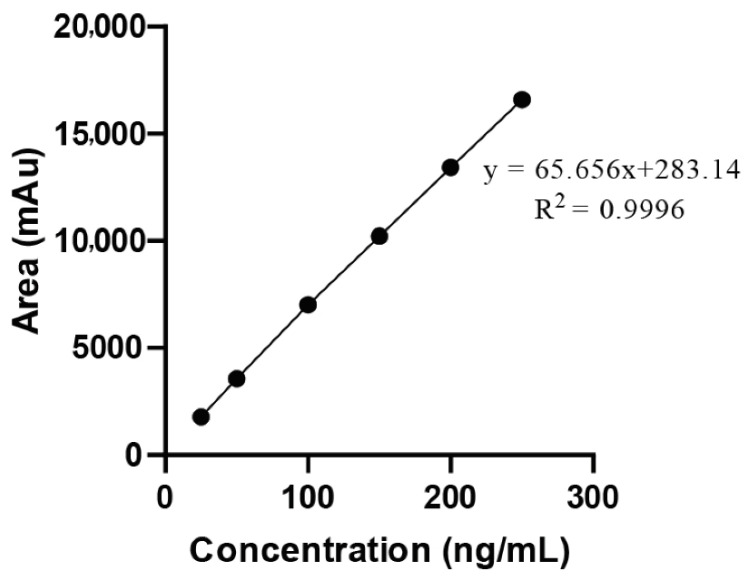
Representing the calibration curve of XH in plasma along with IS.

**Table 1 molecules-27-07117-t001:** Results of matrix effect.

Level	Actual Conc. of XH (ng/mL)	Amount of Drug Recovered in Plasma Sample (ng/mL), (n = 6)	Recovery in Plasma (%)	% RSD
LLOQ	25.73	25.87 ± 0.079	100.50	0.307
HQC	180	175.0 ± 0.18	97.22	0.102

**Table 2 molecules-27-07117-t002:** Accuracy results for XH in plasma.

Level	Actual Conc. of XH (ng/mL)	Amount of Drug Recovered in Plasma Sample (ng/mL), (n = 6)	Recovery in Plasma (%)	%RSD
LLOQ	25.73	24.6 ± 0.05	95.60	0.203
LQC	120	120.9 ± 0.02	100.75	0.556
MQC	150	153.1 ± 0.08	102.00	1.069
HQC	180	179.0 ± 0.09	99.40	1.111

**Table 3 molecules-27-07117-t003:** Interday, intraday, interanalyst precision results for XH in plasma.

Levels	Concentration (ng/mL)	Parameters									
		Interday Precision (Repeatability) (Mean Area ± SD) (n = 6)	%RSD	Interanalyst (Mean Area ± SD) (n = 6)			%RSD	Intraday Precision (Mean Area ± SD) (n = 6)			%RSD
				Analyst 1	Analyst 2	Analyst 3		Day 1	Day 2	Day 3	
LLOQ	25.73	1779.20 ± 15.01	0.84	1786.41 ± 16.01			0.89	1775.95 ± 16.95			0.95
LQC	120	9567.10 ± 38.65	0.93	9749.33 ± 57.56			0.59	9617.05 ± 28.84			0.35
MQC	150	10,629.0 ± 52.11	0.95	10,710.33 ± 76.64			0.72	10,584.89 ± 37.04			0.35
HQC	180	11,425.3 ± 140.17	1.09	11,620.33 ± 73.60			0.63	11,526.50 ± 114.69			0.98

**Table 4 molecules-27-07117-t004:** System suitability results for XH.

Parameter	Value	Limit
Tailing factor	1.187	<2
Theoretical plate	4160.691	>2000
HETP	36.052	Depends on theoretical plate

**Table 5 molecules-27-07117-t005:** Freeze thaw stability for plasma samples of XH.

Actual Concentration of XH (ng/mL)	Mean Area ± SD (n = 3)	% RSD	Actual Amount of XH Recovered in Plasma (ng/mL)	% Recovery
Cycle 1				
LQC	8183.00 ± 60.46	0.73	119.52	99.52
MQC	9999.00 ± 17.04	0.17	147.54	98.34
HQC	12,022.67 ± 54.10	0.44	178.63	99.27
Cycle 2				
LQC	8205.33 ± 108.32	1.32	119.84	99.93
MQC	10,121.00 ± 24.12	0.23	149.31	99.58
HQC	12,019.67 ± 75.21	0.62	178.68	99.25
Cycle 3				
LQC	8226.00 ± 15.82	1.88	120.27	100.13
MQC	10,043.33 ± 38.87	0.38	148.26	98.87
HQC	12,089.33 ± 40.45	0.33	179.78	99.85

**Table 6 molecules-27-07117-t006:** Short term stability for plasma samples of XH.

Actual Concentration of XH (ng/mL)	Mean Area ± SD	% RSD	Actual Amount of XH Recovered in Plasma (ng/mL)	% Recovery
1 h				
LQC	8231.66 ± 90.63	1.11	120.30	100.25
MQC	10,125.33 ± 85.64	0.85	149.46	99.64
HQC	12,114.00 ± 83.83	0.69	180.08	100.04
2nd h				
LQC	8190.66 ± 81.32	0.99	119.67	99.72
MQC	10,081.00 ± 124.49	1.23	148.78	99.18
HQC	12,161.00 ± 94.32	0.78	180.81	100.45
3rd h				
LQC	8124.00 ± 38.19	0.47	118.64	98.86
MQC	10,158.67 ± 44.73	0.44	149.97	99.98
HQC	12,094.33 ± 137.55	1.14	179.78	99.88

**Table 7 molecules-27-07117-t007:** Long term stability for plasma samples of XH.

Actual Concentration of XH (ng/mL)	Mean Area ± SD	% RSD	Actual Amount of XH Recovered in Plasma (ng/mL)	% Recovery
Week 1				
LQC	8131.67 ± 110.21	1.36	118.76	98.92
MQC	10,092.00 ± 90.80	0.89	148.94	99.24
HQC	12,080.67 ± 37.20	0.31	179.57	99.75
Week 2				
LQC	8165.00 ± 112.21	1.38	119.27	99.38
MQC	10,158.67 ± 44.73	0.44	149.97	99.97
HQC	12,147.33 ± 77.35	0.64	180.60	100.35
Week 3				
LQC	8185.00 ± 66.67	0.82	119.58	99.63
MQC	10,042.00 ± 72.84	0.73	148.17	98.78
HQC	12,064.00 ± 75.90	0.63	179.31	99.62

## Data Availability

Not applicable.

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
