# Peer review of "Bioanalytical Method Development, Validation and Stability Assessment of Xanthohumol in Rat Plasma"

_molecules, 2022, doi:10.3390/molecules27207117_

Round 1

Reviewer 1 Report

The manuscript may be accepted after suitable revision. The revised version should address the following comments:

 1- Linearity range (50-250 ng/mL) is not wide enough for a bioanalytical HPLC method. Usually, ranges for HPLC methods used in bioanalysis should have at least one or more orders of magnitude. Also, how come the LOD and LOQ values are by far below the lower concentration in linearity range? Please revise.

2- I guess Figures 1 and 2a are the same, so one of them can be deleted. 

3- A figure should be added showing chemical structures of the drug (Xanthohumol) and the IS (curcumin).

4- Nothing was mentioned under the Results and Discussion section about development and optimization of the method regarding choice of column, mobile phase, detection wavelength, .. etc., and preparation of plasma samples Authors should discuss these points with some details.

5- References and literature review parts are poor. Relevant work including published HPLC methods for xanthohumol should be checked, cited and authors should compare their work to those related published methods.

Author Response

Respected Editor (Molecules Journal),

First of all, we would like to thank the editorial team for the opportunity and we are grateful for your valuable comments in this article. We would also like to express our sincere thanks to the learned reviewers for their sagacious suggestions. We went through the comments and added the required information. Please note that the changes are track change version.

If still any further action is required, then please let us know. We will be happy to address that also.

Reviewer #1

Comment 1 : Linearity range (50-250 ng/mL) is not wide enough for a bioanalytical HPLC method. Usually, ranges for HPLC methods used in bioanalysis should have at least one or more orders of magnitude. Also, how come the LOD and LOQ values are by far below the lower concentration in linearity range? Please revise.  

Response: In the present study the calibration curve was constructed in the range of 50 ng/mL to 250 ng/mL, the range of the calibration is in nanogram level which is also appropriate. There was a typographical mistake in LOD and LOQ in the manuscript. The revised LOD, LOQ values have been inserted in the manuscript (Lines: 168-171). We also inserted the LLOQ values for construction of calibration curve asper ICH M10 guidelines even though, the R2 value is highly significant that confined the selected range is appropriate. The revised calibration curve was inserted in the manuscript (Lines: 142-144; Figure 3)

Comment 2: I guess Figures 1 and 2a are the same, so one of them can be deleted.

Response:  As per the suggestion Figure 1  is deleted in the manuscript.

Comment 3: A figure should be added showing chemical structures of the drug (Xanthohumol) and the IS (curcumin)

Response:  As per suggestion chemical structures of Xanthohumol and curcumin was placed in the manuscript (Lines: 68-69; Figure 1).

Comments 4: Nothing was mentioned under the Results and Discussion section about development and optimization of the method regarding choice of column, mobile phase, detection wavelength, etc. and preparation of plasma samples. Authors should discuss these points with some details.

Response: Development and optimization of the analytical method was written in results and discussion section (Lines: 112-125). Preparation of of plasma samples were clearly explained under materials and methods section (Lines: 230-253).  

Comment 5: References and literature review parts are poor. Relevant work including published HPLC methods for Xanthohumol should be checked, cited and authors should compare their work to those related published methods.

Response:  As per suggestion reference and literature part has revised in the introduction section.

Reviewer 2 Report

Authors presented a Bioanalytical Method Development, Validation and Stability Assessment of Xanthohumol in Rat Plasma. Validation is an essential part of the bioanalytical method and it should be performed following a related guideline. The authors mentioned that the validation of the method was performed as per ICH M10 guidelines. However, more parameters are not compatible with the requirements of this guideline as example:

-          The matrix Effect, Carry-over and Dilution Integrity were not studied

-          The calibration curve was constructed in the range of 50 – 250 ng/ml and according to the applied guidelines the lowest point in the calibration curve should be the LLOQ (0.0139 ng/ml).

-          According to the guidelines mentioned before, the method validation, the QCs should be prepared at a minimum of 4 concentration levels, the (LLOQ, low QC MQC and High QC. In the present work LLOQ is not applied and they used only three points.  

-          The Low QC sample should be, within three times of LLOQ. In the present work, the LLOQ is 0.0139 ng/ml while the Low QC is 120 ng/m L.

-          No details of the thawing process is mentioned (time between the thawing cycles).

There are many notes for this study and as the validation is the most essential process of the bioanalytical method, in my opinion manuscript cannot be accepted  

Author Response

Respected Editor (Molecules Journal),

First of all, we would like to thank the editorial team for the opportunity and we are grateful for your valuable comments in this article. We would also like to express our sincere thanks to the learned reviewers for their sagacious suggestions. We went through the comments and added the required information. Please note that the changes are track change version.

If still any further action is required, then please let us know. We will be happy to address that also.

Reviewer #2

Comment 1 : Matrix Effect, Carry-over and Dilution integrity were not studied  

Response: As per the valuable comment, we have performed as per ICH M10 guidelines matrix effect (Lines: 146-153, 290-293 ) carry-over (Lines: 172-176, 322-327) and dilution integrity (Lines: 172-176, 328-331). The same has been inserted in to the manuscript.

Comment 2: The calibration curve was constructed in the range of 50-250ng/mL and according to the applied guidelines the lowest point in the calibration curve should be the LLOQ (0.0139 ng/mL).

Response:  The calibration curve was reconstructed by adding LLOQ (8 ng/mL) as per ICH M10 guidelines. Revised LOQ and LOD values has placed (Lines: 168-171). There was a typographical error of LOD and LOQ which is corrected in the manuscript.

Comment 3: According to the guidelines mentioned before, the method validation, the QCs should be prepare at a minium of 4 concentration levels, the (LLOQ. Low QC, MQC and High QC. In the present work LLOQ is not applied and they used only three points.

Response: According to ICH M10 guidelines LLOQ was added and performed the validation.   

Comments 4: The Low QC sample should be, within three times of LLOQ. In the present work, the LLOQ is 0.0139 ng/mL while the low QC is 120 ng/mL.

Response:  As per the guidelines it is not mandate to take the points exactly three times greater than LLOQ because there will not be any change in the linearity . According to the revised LOD and LOQ the comment has satisfied.

Comment 5: No details of the thawing process is mentioned (time between the thawing cycles).

Response:  Clearly the details of thawing process was mentioned in the manuscript (Lines: 337-354)

Round 2

Reviewer 1 Report

Authors responded to all comments. Manuscript can be accepted.

Author Response

Respected Editor (Molecules Journal),

First of all, we would like to thank the editorial team for one again giving the opportunity to revise our manuscript and we are grateful for your valuable comments in this article. We would also like to express our sincere thanks to the learned reviewers for their sagacious suggestions. We went through the comments and added the required information. Please note that the changes are track change version.

If still any further action is required, then please let us know. We will be happy to address that also.

Reviewer #2

Comment 1 : Authors respond to all comments. The LLOQ according to the guidelines must be applied in accuracy and precision only, however authors added to in all results of validation. In the tables LLOQ is 8.0 ng/mL and in page 6-line authors mentioned that the LOD and LOQ for XH in plasma samples were found to be 8.4ng/mL and 25.73 ng/mL respectively please clarify what is 25.73 ng/mL.  

Response: Firstly, I would like to thank reviewers for providing the valuable suggestions and comments that improve the strength of the manuscript. As per the suggestion and comment we revised the manuscript. LLOQ value is updated. As the mentioned LLOQ in previous revision is a typographical mistake. Actually, 25.73 ng/mL is the LLOQ and 8.4 ng/mL is LOD. Manuscript was revised accordingly and updated the same in manuscript. As per the guidelines during bioanalytical validation LOQ is termed as LLOQ.

Reviewer 2 Report

Authors respond to all comments. The LLOQ according to the guidelines must be applied in accuracy and precision only, however authors added to in all results of validation. In the tables LLOQ is 8.0 ng/mL and in page 6-line authors mentioned that “The LOD and LOQ for XH in plasma samples were found to be 8.4 ng/mL and 25.73 ng/mL, respectively’’. Please clarify what is 25.73 ng/mL

Author Response

(The authors gave the same response as above.)
